# Microbial Composition of a Traditional Fermented Wheat Preparation—Nishasta and Its Role in the Amelioration of Retinoic Acid-Induced Osteoporosis in Rats

Aayeena Altaf [1], Naila H. Alkefai [2], Bibhu Prasad Panda [3,4,*], Zakiya Usmani [4], Saima Amin [5] and Showkat R. Mir [4,6,*]

[1] Department of Food Technology, School of Interdisciplinary Science and Technology, Jamia Hamdard, New Delhi 110062, India; ayeena598@gmail.com
[2] Department of Pharmaceutical Chemistry, College of Pharmacy, University of Hafr al-Batin, Hafr Al-Batin 39524, Saudi Arabia; nailahsn2@gmail.com
[3] Microbial & Pharmaceutical Biotechnology Laboratory, School of Pharmaceutical Education & Research, Jamia Hamdard, PO Hamdard Nagar, New Delhi 110062, India
[4] Department of Pharmacognosy & Phytochemistry, School of Pharmaceutical Education & Research, Jamia Hamdard, PO Hamdard Nagar, New Delhi 110062, India; zakiyausmani9@gmail.com
[5] Department of Pharmaceutics, School of Pharmaceutical Education & Research, Jamia Hamdard, PO Hamdard Nagar, New Delhi 110062, India; samin@jamiahamdard.ac.in
[6] Phyto-Pharmaceuticals Research Laboratory, School of Pharmaceutical Education & Research, Jamia Hamdard, PO Hamdard Nagar, New Delhi 110062, India
[*] Correspondence: bppanda@jamiahamdard.ac.in or bibhu_panda31@rediffmail.com (B.P.P.); showkatrmir@gmail.com or srmir@jamiahamdard.ac.in (S.R.M.)

**Abstract:** Fermented foods have a long history of human use. The purpose of this study was to characterize the microbial composition of a traditional fermented wheat preparation—Nishasta— and to explore its effect in retinoic acid-induced osteoporosis in Wistar rats. The sample was suspended in sterile water (10% *w/v*), mixed thoroughly, filtered, and gradually diluted. Aliquots of dilutions were cultured in MRS (DeMan–Rogosa–Sharpe) medium, and colonies with similar morphologies were subjected to DNA extraction. The 16S rRNA gene of the isolates was amplified by polymerase chain reaction, checked by agarose gel electrophoresis, and finally identified by sequencing. Anti-osteoporosis screening of Nishasta was carried out in female Wistar rats using retinoic acid as an inducer (70 mg/kg, p.o. once a day for 14 days). Its effect on bone health parameters was determined. The bone metabolism markers such as hydroxyproline (HOP), tartrate-resistant acid phosphatase (TRACP), and alkaline phosphatase (ALP) were evaluated. The results of microbial characterization revealed the presence of ten clones of *Lactobacillus plantarum* in the fermented preparation with *L. plantarum* NF3 as the predominant strain. The average microbial count was $2.4 \times 10^3$ CFU/g. Retinoic acid administration led to a marked disorder of various bone health markers in rats. It also increased the levels of urine calcium and phosphorus, indicating increased bone destruction. Treatment with fermented wheat (at 200, 100, and 50 mg/kg doses, p.o. daily for 42 days after the induction of osteoporosis) improved bone mineral density in a dose-dependent manner. It also improved the bone microstructure and reduced the levels of ALP, TRACP, and HOP. Micro-CT revealed that it reduced trabecular separation and increased the percent bone volume, trabecular numbers, trabecular thickness, and bone mineral density in the rats. The results showed that the fermented wheat promoted bone formation and prevented bone resorption. Our findings clearly established the effectiveness of Nishasta against osteoporosis in Wistar rats that can be partly attributed to the improved gut calcium absorption and microbiota composition.

**Keywords:** fermented food; wheat; Nishasta; bone health; bone mineral density; *Lactobacillus plantarum*

## 1. Introduction

Osteoporosis is a progressive systemic disorder that is characterized by a loss of bone mass and disruption of bone microarchitecture that lead to bone fragility and increased risk of fractures. The incidence of osteoporosis is higher in elderly people. It is more common in women than in men. There are two types of osteoporosis. Type I or primary osteoporosis is characterized by rapid bone loss that occurs after the onset of menopause triggered by lower levels of estrogen. This in turn results in more bone resorption than bone formation. Type II or secondary osteoporosis occurs due the prolonged deficiency of calcium levels in the body. It mainly occurs with aging and is also called senile osteoporosis. It also occurs due to increased parathormone activity resulting a decline in the formation of bones [1]. Osteoporosis is also caused by the use of drugs such as heparin, gluco-corticosteroids, and chemotherapeutic agents, by smoking or by the use of alcohol. Bones weaken to such an extent that a break can occur with minor stress or spontaneously. Prolonged pain and decreased ability to carry out normal activities are the symptoms of broken bones. There are many potential bone-forming drugs that are used against osteoporosis but their use is often associated with some serious side effects and does not decrease the risk of bone fractures significantly [2]. Thus, there is a serious need to look for some alternatives such as food supplements that can decrease bone loss in the elderly and in women.

Fermented foods have emerged as one of the important areas of research in the field of food sciences. Diet and bone mass density are significantly correlated [3,4]. Several studies indicate that the consumption of fermented foods has proved beneficial in maintaining bone health in experimental animals [5,6]. Fermented food supplements have been found to be effective to ward off other deficiency diseases [7–10]. Different in-vivo and in-vitro studies have suggested that cereal-based fermented products improve bone health through the stimulation of bone formation and inhibition of bone resorption. Fermented soy products have been reported to promote the synthesis of osteoblast proteins and proliferation of osteoblasts that in turn improve bone consolidation. Increased bone density may be attributed to enhanced calcium and phosphorus contents in the bone tissues [11,12].

This study has its basis on a traditionally used fermented wheat preparation called Nishasta in Kashmir Province of Jammu and Kashmir, India. It is widely used as a general tonic particularly in postpartum and postmenopausal women. Nishasta is conventionally prepared by soaking and fermenting wheat grains in water for about one month to allow natural fermentation to take place. The water is changed every week to allow fresh microbial growth. This is followed by crushing the grains and filtering the residue through a muslin cloth and then drying [13]. The main objectives of the study were (a) to characterize the microbial composition of Nishasta and (b) to evaluate its effectiveness in improving the bone health parameters in retinoic acid-induced osteoporotic rats. In this study, retinoic acid was used to establish a rat model for osteoporosis, and the preventive effect of Nishasta was investigated using this model.

## 2. Material and Methods

### 2.1. Materials

MRS agar (cat. M641) was purchased from HiMedia, Mumbai, India. Retinoic acid (cat. R2625) was obtained from Sigma Aldrich, Mumbai, India. Sodium alendronate was procured from Apex Healthcare Limited, Ankleshwar, Gujarat, India. Commercial kits for the determination of calcium (cat. MAK022), phosphorus (cat. MAK308), hydroxyproline (cat. MAK357), tartrate resistant acid phosphatase (cat. P13686), and alkaline phosphatase (cat. SCR004) were purchased from Sigma Aldrich, Mumbai, India. All other chemicals and reagents of analytical grade were purchased from SD Fine Chemicals, New Delhi, India.

### 2.2. Method of Fermentation and Analysis

Wheat (*Triticum aestivum* variety Shalimar Wheat 1) was procured from the Sher-e-Kashmir Agriculture University of Science and Technology, Srinagar, Jammu and Kashmir, India, under the registration No. Au/MRCFC/FS/332. Wheat grains (1000 g) were sorted

and washed with water. The grains were then soaked and incubated with water (1:3 *w/v*) at room temperature (25 ± 1 °C) for a time period of four weeks. Water was replaced every week in order to allow fresh microbial growth to affect natural fermentation. At the end of the four weeks, the water was strained and the softened mass was ground in an electrical grinder at 2000 rpm for one minute. The mixture was filtered through a muslin cloth to remove germs and bran. The resulting filtrate was centrifuged at 3000 rpm for 10 min. The sediment was spread in shallow trays and dried at 45 °C. Three batches of Nishasta were prepared using the same procedure. The methodology followed for preparing Nishasta is conventionally used to prepare other traditional Indian fermented products [14]. Dried Nishasta powder was subjected to proximate analysis using AOAC methods [15].

### 2.3. Isolation and Molecular Characterization of Microbes

Nishasta samples were suspended in sterile water (10% *w/v*), mixed thoroughly, filtered, and gradually diluted. Dilutions up to $10^{-5}$ were prepared and applied to MRS (DeMan–Rogosa–Sharpe) agar plates in duplicate, and all plates were incubated at 37 °C for 24–48 h. The culture characteristics and morphology of colonies were observed, and colonies with similar patterns were selected. Pure colonies were inoculated with MRS liquid medium (5 mL) and cultured at 37 °C for 18–24 h. The above 1 mL medium was centrifuged at 12,000 rpm for 1 min, the supernatant was discarded, and 200–500 μL of sterile normal saline was added. Genomic DNA was isolated from a 0.1 mL sample of overnight cultures grown in MRS liquid medium. gDNA was isolated from all bacterial isolates and used as template for PCR. The primers used for the amplification of parts of 16S rRNA were forward: 5′-AGAGTTTGATCMTGGCTCAG-3′; reverse: 5′-TACGGYTACCTTGTTACGACTT-3′. The 16S rRNA gene of the samples was amplified by PCR polymerase, and the product was checked by agarose gel electrophoresis. The PCR product was sequenced bi-directionally, and then the data were aligned and analyzed to identify the sample. The PCR product size was ~1.5 kb, and a 500-base pair DNA ladder was used to estimate the size of DNA fragments. Sequencing was performed by using an ABI 3500 Genetic Analyzer in which a POP-7 Polymer and 50 cm Capillary Array were used. A BigDye® Terminator version 3.1 Cycle Sequencing Kit (Thermo Fisher Scientific, Wilmington, DE, USA) was used in combination with an Applied Biosystem Micro Amp Optical 96-Well Reaction plate. The sequencing reaction mix volume was 10 μL, which contained 4 μL of ready reaction mix, 1 μL of the template (100 ng/μL), and 2 μL of primer (10 pmol/μL) with 3 μL of Milli Q water. The PCR for sequencing was completed in 25 cycles. Initial denaturation was performed for 5 min at 96 °C followed by denaturation for 30 s at 96 °C. Hybridization and elongation were performed at 50 °C for 30 s and 60 °C for 1.30 min, respectively. Phylogenetic Tree Builder was used to generate a phylogeny tree from 16S rRNA sequences aligned with System Aligner. A distance matrix was generated using the Jukes–Cantor corrected distance model. While generating the distance matrix, only alignment model position was used, alignment inserts were ignored, and the minimum comparable position was 200. Bootstrap analysis was used to determine the stability of the phylogenetic tree. The process was repeated 100 times, and a majority consensus tree was displayed showing the number of times (or percentage) that a particular group was on each side of a branch without subgrouping.

### 2.4. Animals and Experimental Design

The general guidelines of the Committee for the Purpose of Control and Supervision of Experiments on Animals (CPCSEA) for the care and use of laboratory animals were strictly adhered to during the study. The study protocol was approved by the Institutional Animal Ethics Committee (IAEC), Jamia Hamdard (Registration No. 173/GO/RE/S/2000/CPSEA) under the proposal number 1597. Thirty-six female Wistar rats aged 90 days with a body weight range of 180–200 g were supplied by the Central Animal House Facility, Jamia Hamdard, New Delhi. Six rats were housed in each polypropylene cage placed in a room with a 12-h/12-h light–dark cycle, 50 ± 10% relative humidity, and 23 ± 1 °C temperature.

The animals had free access to water and diet. Animals were acclimatized for one week before the experiment.

Before the administration of samples, the rats were fasted for 12 h. A total of 36 Wistar rats were used for the study, of which 30 rats were administered retinoic acid (70 mg/kg, p.o.) once a day for 14 days in order to induce osteoporosis. After the induction of osteoporosis, the rats were randomly distributed into five groups of six rats each. Rats receiving retinoic acid acted as the model ($n = 6$). The positive control rats were administered sodium alendronate (3.75 mg/kg, p.o.) once daily for the next 42 days. The treatment groups were administered 200, 150, and 100 mg/kg of fermented wheat, once daily for the next 42 days by oral gavage. These constituted high-, medium-, and low-dose fermented wheat treated groups and were indicated as HD-FW, MD-FW, and LD-FW, respectively. Doses administered were selected and adjusted according to the body weight of the rats on a daily basis [16]. All test samples were prepared using 0.5% CMC in distilled water as the vehicle. The remaining six rats served as healthy control animals and were administered an equivalent amount of vehicle orally (1 mL/kg body weight). The blood samples from rats were collected at pre-determined intervals (0, 14 days and 24 h after the last dose of the treatment schedules) for biochemical analysis. Urine samples were also collected at these intervals for the estimation of calcium and phosphorus in urine. Finally, the rats were sacrificed and their hind legs were dissected. The abdomen was cut open to harvest kidneys, liver, pancreas, spleen, lungs, and heart. The organs were fixed with 10% formalin solution for 12 h and embedded in paraffin. Five micron (5 μm) cryostat sections were stained with hematoxylin and eosin dyes. The sections were examined under a light microscope at power magnifications of 40× and 10× to study changes in the tissue architecture and for capturing photomicrographs.

*2.5. Measurement of Bone Quality Parameters*

The femoral bones were dissected from the animals, and the adhering tissue was cleaned with the help of a sterile surgical blade. The femur bone was weighed to obtain the bone weight index (g/100 g body weight). The diameter and length of bones were measured by using stainless steel Vernier calipers (Fowler 52-085-040). Bones were dried at 110 °C for about 2 h and the weight of bones recorded. The trabecular volume, thickness, number, and separation of femoral bones were calculated [17]. The femur samples were dipped in phosphate buffer solution at room temperature, and BMD was measured by a dual energy X-ray bone densitometer apparatus (Discovery A model, Hologic software version 12.5, SN = 08E0204θ and Skyscan 1076, version 3.1 Aartselaar, Belgium) at a nominal resolution used in the case of small animals and was expressed as the bone mineral content per unit area. The X-ray source was set to 70 kV and 100 mA [18]. For the determination of the ash content, the femur bone was dehydrated followed by incineration in a muffle furnace at 750 °C for 9 h. The ash was extracted with 6N HCL, and the contents of phosphorus and calcium were determined [19].

*2.6. Biochemical Estimations*

Blood samples were collected at pre-determined intervals as mentioned above, and serum was obtained by centrifugation at 3000 rpm for 10 min at 4 °C. Calcium and phosphorus levels in the serum were measured by atomic absorption spectrometry after sample preparation as per the manufacturer's instructions. Urine samples were centrifuged at 3000 rpm for 10 min, the supernatant was collected for subsequent analysis, and the concentration of urinary calcium and phosphorus was estimated. The sample preparation was carried out as per the assay kits' instructions. Serum TRACP and ALP activities were measured by spectrophotometry. For their estimation, the blood samples were centrifuged at 1000 rpm for 20 min. The supernatant was collected and incubated at 37 °C. The absorbance was recorded at 450 nm [20]. The values for TRACP and ALP were expressed in U/L and mmol/L, respectively. For HOP estimation, the urine samples from rats were treated with hydrochloric acid and incubated for 3 h at 120 °C. This was followed by adding

5 mg of activated charcoal, mixing on a vortex mixer, and centrifugation at $10,000 \times g$ for five minutes. The supernatant was collected, and the HOP level was determined by recording the absorbance at 562 nm, which was reported as mM/mg [21].

### 2.7. Statistical Analysis

All determinations were carried out in triplicate, and the data were expressed as the mean value ± SD. The results were analyzed by ANOVA using Student's *t*-test procedure to determine the level of significance; *p* values of less than 0.05 were considered to be statistically significant. The analysis was performed using GraphPad Prism software (GraphPad Software Inc., Version 7, Chicago, IL, USA).

## 3. Results

### 3.1. Preparation of Nishasta and Proximate Analysis

Figure 1 depicts the procedure for the preparation of a representative batch of Nishasta. Fermentation of wheat (1000 g) resulted in the production of about 227 ± 6 g of Nishasta. The samples were white to cream colored, amorphous in nature, slightly hygroscopic, and generally in the form of friable lumps.

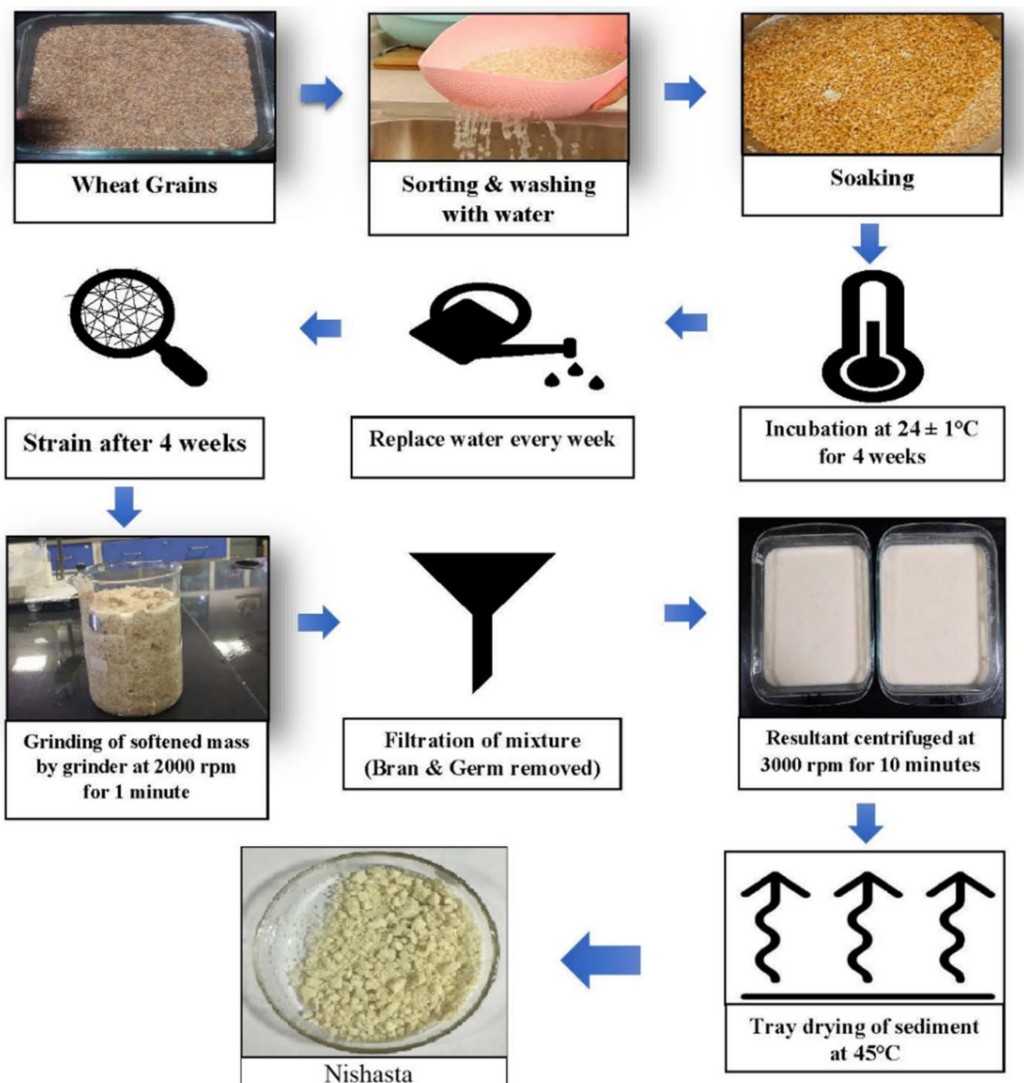

**Figure 1.** Steps involved in the preparation of Nishasta by fermentation of wheat.

The results of the proximate analysis of the fermented wheat compared to unfermented wheat are presented in Table 1. The fermented wheat was characterized by a low pH. Its

1% $w/v$ solution in distilled water was found to be acidic in nature (pH 3.12). The low pH of the samples is due the production of short-chain fatty acid (SCFAs) during the fermentation of wheat. Microbial fermentation and hydrolysis resulted in a reduced fiber content in the fermented samples. The protein and lipid contents were significantly higher in fermented samples compared to the unfermented wheat.

**Table 1.** Proximate analysis of fermented wheat preparation (Nishasta).

| S. No. | Content (g/100 g) | Unfermented Wheat | Fermented Wheat |
|:---:|:---:|:---:|:---:|
| 1 | Protein | $11.73 \pm 0.39$ | $17.23 \pm 0.40$ * |
| 2 | Lipid | $2.43 \pm 0.12$ | $15.26 \pm 0.80$ * |
| 3 | Fiber | $7.19 \pm 0.09$ | $2.54 \pm 0.24$ * |
| 4 | Moisture | $24.37 \pm 0.29$ | $3.86 \pm 1.71$ * |
| 5 | Ash | $6.4 \pm 0.15$ | $6.33 \pm 0.08$ [ns] |

Data expressed as the mean $\pm$ SEM; * indicates $p < 0.001$ and [ns] $p > 0.5$ compared to the unfermented wheat.

### 3.2. Microbiological Analysis

Culturing of inoculates from the dilutions of Nishasta on MRS agar plates resulted in the isolation of distinct, punctiform, and convex colonies. Their color varied from white to buff. The number colonies averaged around 24 when 1 mL of $10^{-2}$ dilution was incubated on MRS agar medium. The total microbial load of the samples averaged $2.4 \times 10^3$ CFU/g. The colonies with similar morphologies were incubated again in MRS liquid medium. Representative isolates were carefully chosen for the extraction of DNA. The 16S rRNA gene of the isolates were amplified by PCR, checked by agarose gel electrophoresis, and finally identified by sequencing. The PCR analysis yielded a single amplicon band at about 1500 bp, as shown in Figure 2. The phylogenetic tree of the strains isolated from the fermented wheat revealed that the majority of the strains grown on MRS agar medium belonged to the *Lactobacillus* genus with *L. plantarum* as the predominant microbe. Large intra-species diversities were observed among the isolates. Overall, 10 different strains of *L. plantarum* with 99% similarity were identified. Their phylogenetic details are presented in Figure 2.

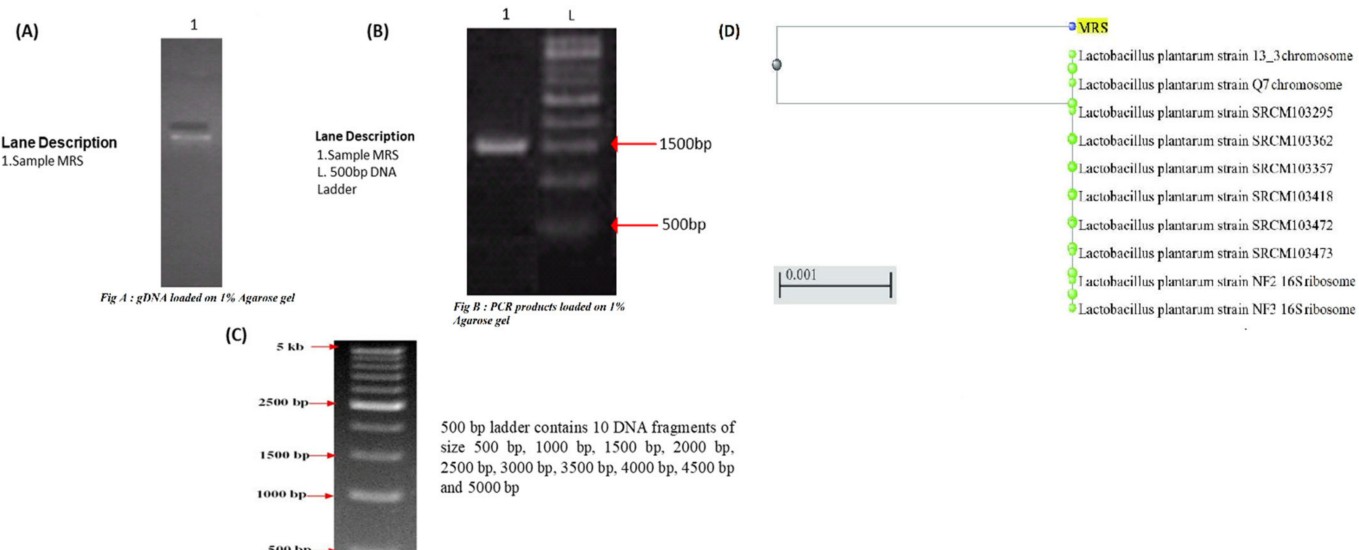

**Figure 2.** (**A**–**C**) Agarose gel electrophoresis of PCR products of the 16S rRNA gene from microbes isolated from Nishasta samples and (**D**) Phylogenetic tree of *Lactobacillus plantarum* isolates.

### 3.3. Effect of Fermented Wheat on Body Weight

The effect of fermented wheat on the body weight of rats was evaluated, and the results are presented in Figure 3. The average body weight of the animals ranged between 180 and 200 g at the start of the study. Oral administration of 75 mg/kg retinoic acid to the rats for two weeks (designated as the model) resulted in a continuous decline in their body weight compared to healthy control rats. Sodium alendronate-treated rats (designated as the positive control, PC) showed a gradual increase in body weight during the treatment. High, medium, and low doses of fermented wheat-treated rats (HD-FW, MD-FW, and LD-FW) also showed a steady increase in body weight commensurate with the dose of the samples tested. Fermented wheat resulted in significant improvements in the body weight of rats treated with high and medium doses ($p < 0.001$) compared to the ones treated with a low dose ($p < 0.01$).

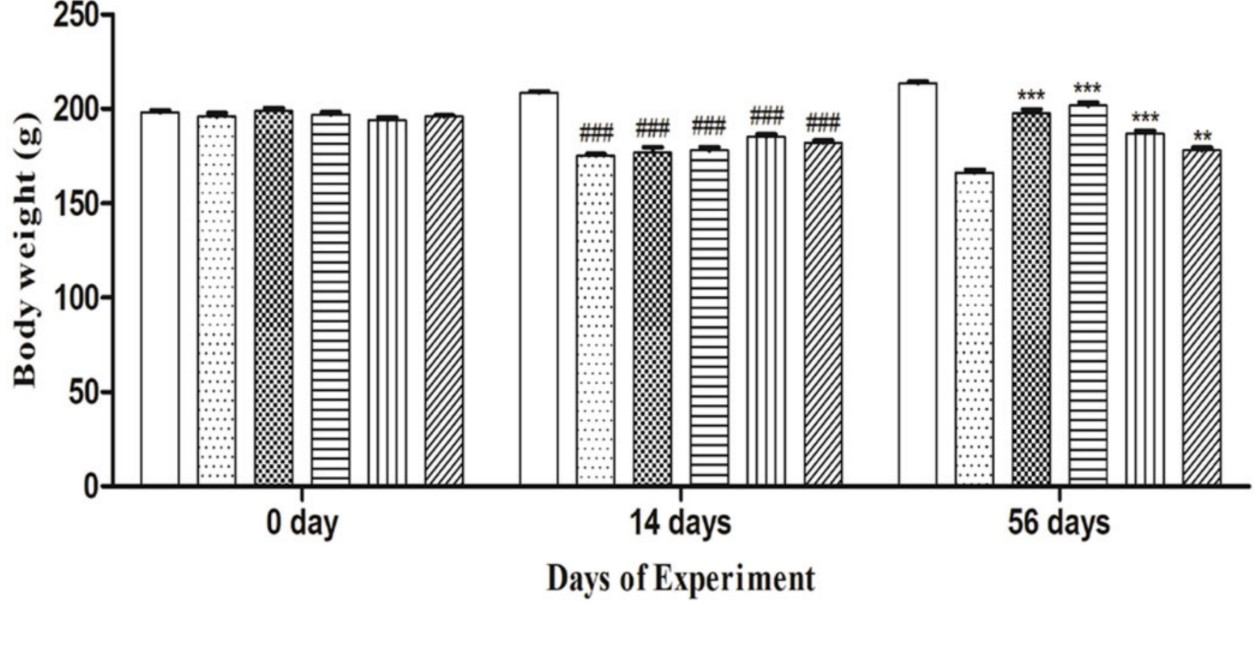

**Figure 3.** Effect of fermented wheat on the body weight of rats. Results are significantly different from the healthy control ([###] $p < 0.001$) and the model ([***] $p < 0.001$, [**] $p < 0.01$).

### 3.4. Effect of Fermented Wheat on Biochemical Indicators

Serum calcium, phosphorus, ALP, and TRACP levels were estimated, and the results are presented in Figure 4. These markers are widely used to assess the effectiveness of a treatment against osteoporosis. On administration of retinoic acid, serum calcium and phosphorus levels decreased significantly compared to the HC group. These results along with increased serum ALP and TRACP levels in retinoic acid-treated animals indicated successful induction of osteoporosis. Treatment with HD-FW, MD-FW, and LD-FW resulted in marked increases in serum calcium and phosphorus in rats compared with the model. Levels of serum ALP and TRACP decreased significantly in HD-FW-, MD-FW-, and LD-FW-treated groups compared with the model. Treatment with sodium alendronate restored the levels of these markers to normal. The results of the effect of fermented wheat administration on the excretion of calcium, phosphorus, and HOP are presented in Figure 5. This clearly showed that after the treatment with fermented wheat, the excretion of calcium, phosphorus, and HOP in urine decreased significantly in treated groups compared with the model.

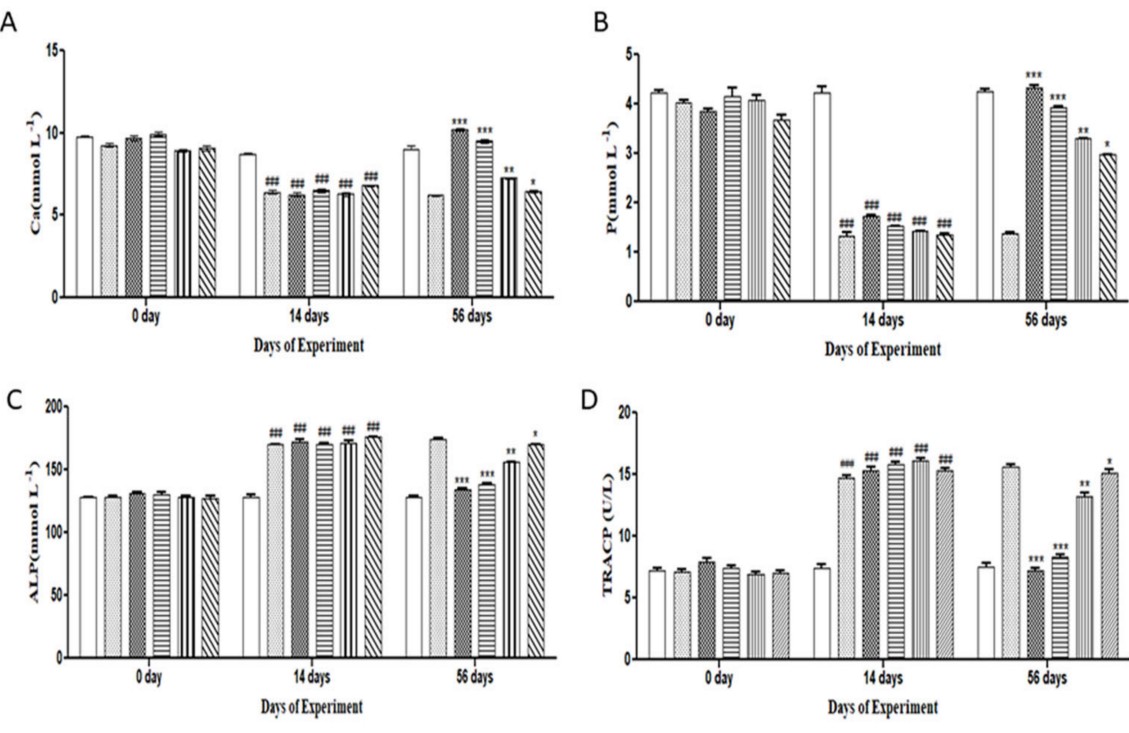

**Figure 4.** Effect of fermented wheat on the blood biochemical parameters of rats. Serum levels of (**A**) Calcium, (**B**) Phosphorus, (**C**) Alkaline phosphatase (ALP), and (**D**) Tartrate-Resistant Acid Phosphatase (TRACP) of rats in each group. Results are significantly different from the healthy control (### $p < 0.001$) and the model (*** $p < 0.001$, ** $p < 0.01$, * $p < 0.05$).

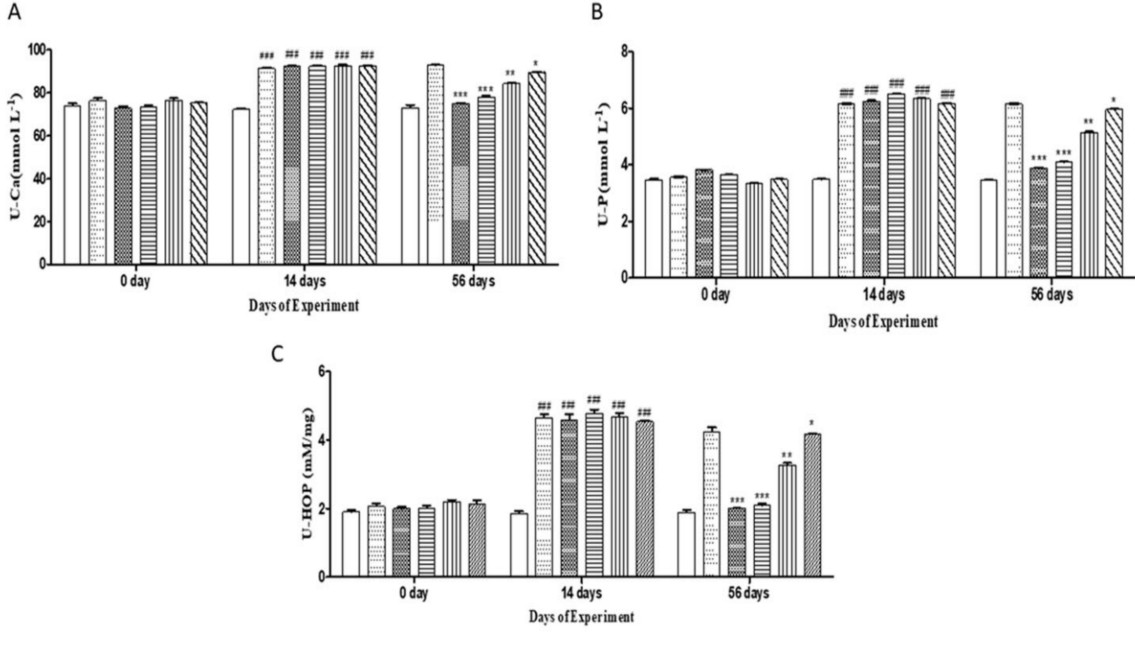

**Figure 5.** Effect of fermented wheat on urine biochemical parameters of rats. Urine levels of (**A**) Calcium, (**B**) Phosphorus, and (**C**) Hydroxyproline (HOP) of rats. Results are significantly different from the healthy control (### $p < 0.001$) and the model (*** $p < 0.001$, ** $p < 0.01$, * $p < 0.05$).

### 3.5. Effect of Fermented Wheat on the Bone Quality of Rats

The outcomes of the effect of administration of fermented wheat on the bone quality parameters are presented in Figure 6. Upon treatment with fermented wheat, the bone weight index was significantly enhanced compared to the model ($p < 0.05$). The index was 0.38 ± 0.01 in HD-FW-treated animals compared to 0.29 ± 0.02 in the model group. The head diameter and length of the femur increased significantly due to treatment with fermented wheat. The effect of HD-FW was particularly comparable to sodium alendronate. The effect of fermented wheat on the ash content of bones was also estimated. The bone ash content in fermented wheat-treated animals improved markedly (34.85 ± 0.07 g/100 g for HD-FW compared to 27.43 ± 0.05 g/100 g for the model). A dose-dependent effect of fermented wheat in improving the bone parameters was also observed. The dimensions of the femur was highest for high dose-treated animals and lowest for low dose-treated rats.

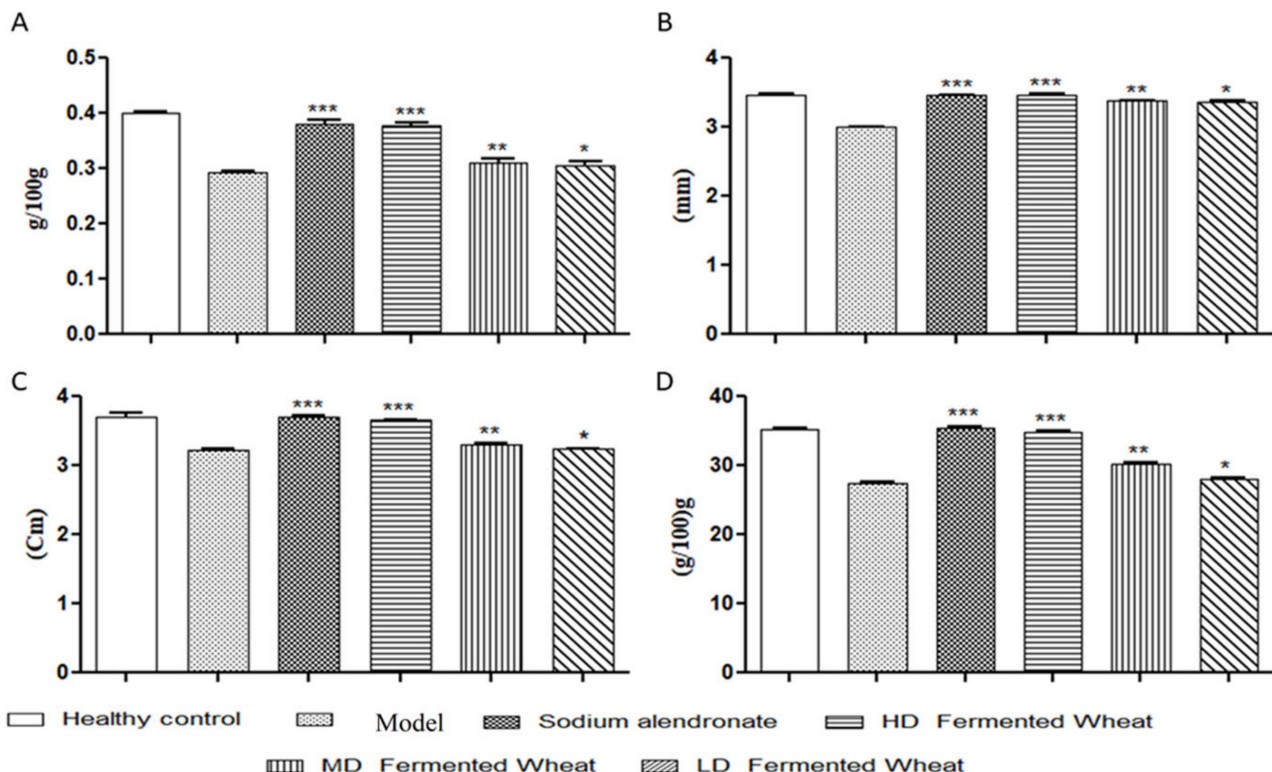

**Figure 6.** Effect of fermented wheat on the bone quality parameters of rats. (**A**) Bone weight index, (**B**) Diameter of femur head, (**C**) Length of femur bone, and (**D**) Ash content of bones. Results are significantly different from the model (*** $p < 0.001$, ** $p < 0.01$, * $p < 0.05$).

### 3.6. Effect of Fermented Wheat on Bone Calcium and Phosphorus Contents

Bone calcium and phosphorus levels were also determined, and the results are depicted in Figure 7. Their enrichment in bones is helpful in osteoporosis. It was observed that after administration of retinoic acid, the bone calcium and phosphorus levels decreased significantly compared to the HC group. Upon treatment with fermented wheat, the bone calcium and phosphorus contents showed a marked increase compared with the model. Treatment with sodium alendronate was also able to restore their contents to normal. Fermented wheat resulted in significant improvements in the bone calcium and phosphorus contents of rats treated with high ($p < 0.001$) and medium doses ($p < 0.01$).

### 3.7. Effect of Fermented Wheat on the Microstructure and BMD of Bones

Retinoic acid administration to Wistar rats caused a marked deterioration of micro-architecture and mineral density of femur and tibia bones (Figures 8 and 9). The results showed decreased percentages of bone volume, trabecular thickness, and trabecular num-

ber and increased trabecular separation compared with the control group. Treatment with HD-FW (200 mg/kg) had a positive effect on bone micro-architecture in retinoic acid-induced osteoporosis in rats. Administration of retinoic acid to rats produced a significant decrease in the BMD of the femur and tibia. Treatment with fermented wheat showed a dose-dependent improvement in BMD compared with the model. Fermented wheat resulted in a significant improvement in the bone parameters of rats treated with a high dose ($p < 0.001$), comparable to that of sodium alendronate.

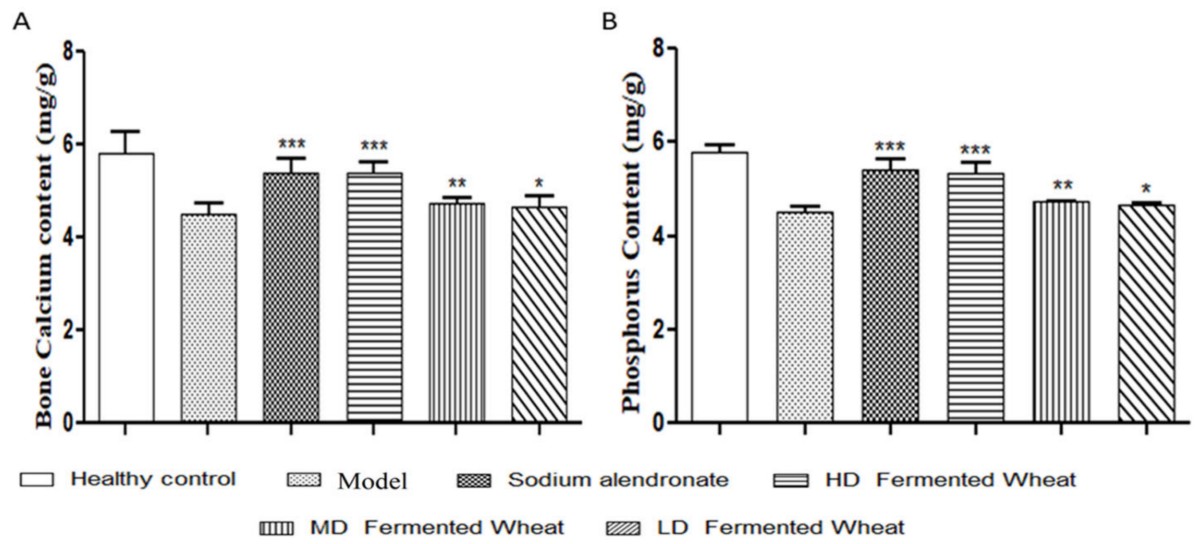

**Figure 7.** Effect of fermented wheat on bone mineral content of rats. (**A**) Calcium and (**B**) Phosphorus contents. Results are significantly different from the model (*** $p < 0.001$, ** $p < 0.01$, * $p < 0.05$).

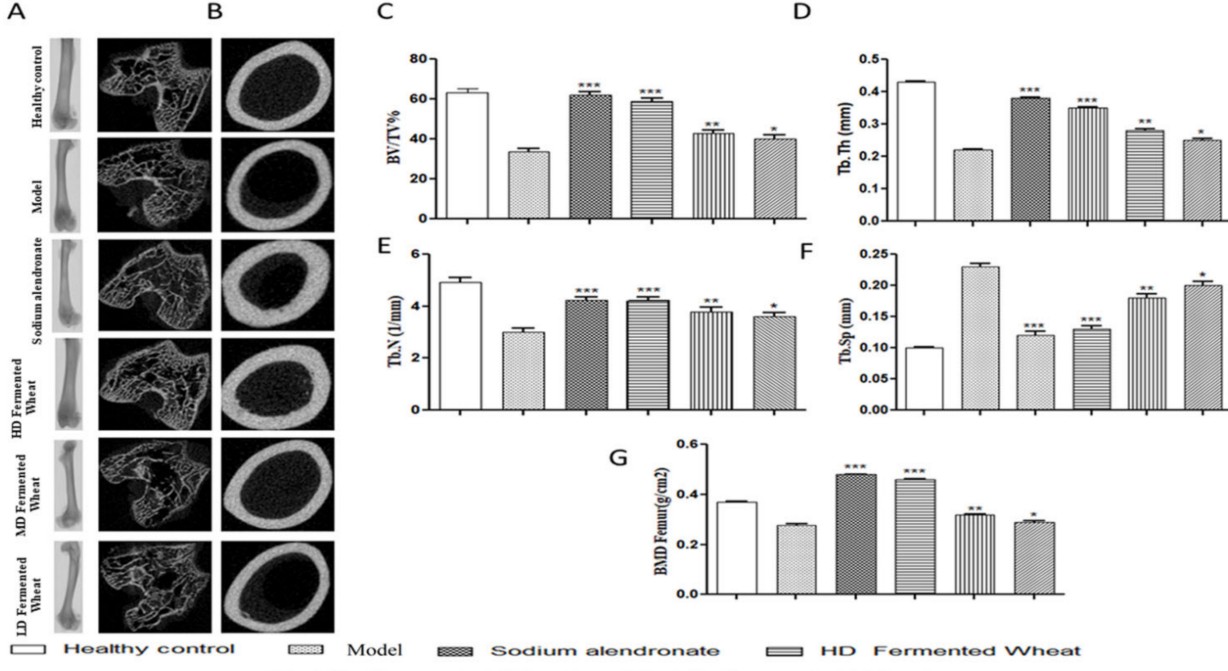

**Figure 8.** Effect of fermented wheat on the femur microstructure in retinoic-induced osteoporotic rats. (**A,B**) Micro-CT detection after the administration of fermented wheat or sodium alendronate. 2D and 3D reconstruction images taken by micro-CT showed reduction in femoral trabeculae and thickness of femur diaphysis in the model. (**C**) Microstructure parameters of BV/TV, (**D**) Tb. Th, (**E**) Tb. N, (**F**) Tb. Sp, and (**G**) BMD of femur improved in fermented wheat-treated rats. Results are significantly different from the model (*** $p < 0.001$, ** $p < 0.01$, * $p < 0.05$).

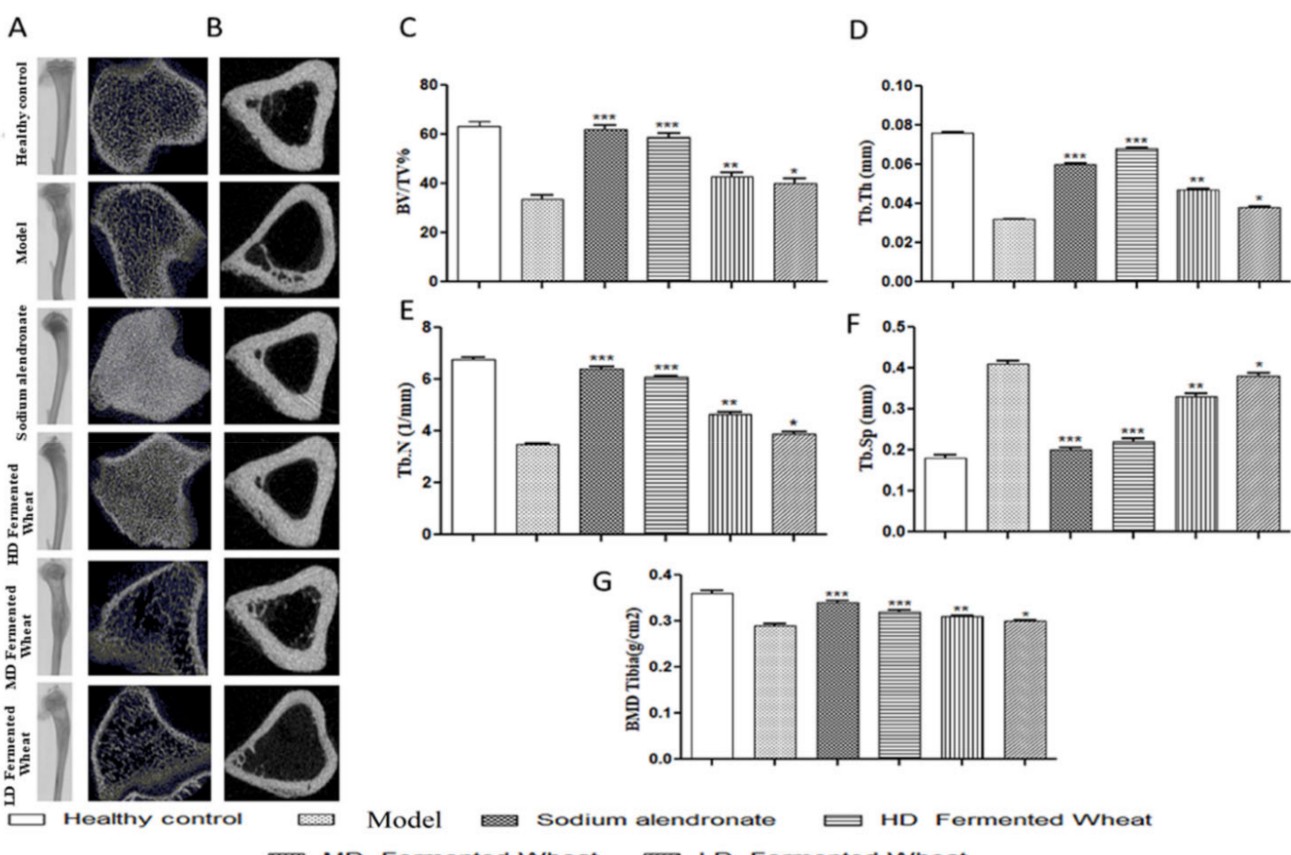

**Figure 9.** Effect of fermented wheat on the tibia microstructure in retinoic-induced osteoporotic rats. (**A**,**B**) Micro-CT detection after the administration of fermented wheat or sodium alendronate. 2D and 3D reconstruction images taken by micro-CT showed reduction in tibia trabeculae and thickness of tibia diaphysis in the model. (**C**) Microstructure parameters of BV/TV, (**D**) Tb. Th, (**E**) Tb. N, (**F**) Tb. Sp, and (**G**) BMD of tibia improved in fermented wheat-treated rats. Results are significantly different from the model (*** $p < 0.001$, ** $p < 0.01$, * $p < 0.05$).

### 3.8. Effect of Fermented Wheat on the Histology of Vital Organs

Macroscopic examination of the sections of vital organs (kidneys, liver, pancreas, spleen, lungs, and heart) of experimental animals showed noticeable damage produced by retinoic acid. The sections of kidney displayed signs of segmental sclerosis and segmentation of glomeruli, indicating cellular injury to mesangium and glomeruli capillaries after the rats were administered retinoic acid. Widening of Bowman's space with hemolysis was also observed. The liver showed hepatic cells with ballooning, distortion of hepatic cell, micro vesicular steatosis (i.e., accumulation of small fat droplets in the hepatocyte cystol), and inflammation. Lungs showed the presence of poly-morphonuclear cells, alveolar thickening, and congestion after administration with retinoic acid. The pancreas sections exhibited decreased numbers of islets and acinar cells. The sections also showed shrunken and distorted islet of Langerhans, vacuolated cytoplasm, and large, darkly stained nuclei, indicating degenerative and necrotic changes. Cellular disintegration and loss of nuclei material were observed in spleen sections. Cardiac muscle showed ruptured and atrophic muscle fibers along with hemorrhage in cardiac muscles. Increased separation of muscle fibers shown by fading of the stain indicated muscle degeneration in rats administered retinoic acid. The fermented wheat treatment was successful in reversing most of the pathological changes induced by retinoic acid in the vital organs. The effects of fermented wheat on the histology of the vital organs are presented as photomicrographs in the Supplementary Data (Figure S1).

## 4. Discussion

Fermented foods and beverages have been the part of the human diet for a long period of time. Fermentation is used for food preservation and/or to improve taste. It also results in enhancement of the content of food constituents such as vitamins, proteins, essential fatty acids, and amino acids. A number of fermented foods are derived from milk, cereals, soybeans, fruits, vegetables, legumes, and fish. The unique flavor and texture of fermented foods are due to the presence and activity of microorganisms, commonly referred to as probiotics. Wheat constitutes a vital portion of staple food for a large part of the global population. It is consumed as a whole grain, soft wheat or as flour. It is also utilized in the preparation of fermented foods such as Injera and Khambir or beverages such as Boza, Kishk, and Trahana in different regions of the world [9]. India also has a long history of the use of wheat-derived fermented foods such as Dosa, Dhokla, Idli, Khaman, Nishasta, Seera, etc.

Nishasta is a traditional fermented wheat preparation used in Kashmir Province of Jammu and Kashmir and other Himalayan regions in the north of India. It is a reputed general tonic used particularly by postpartum and postmenopausal women. There are two purported methods for the preparation of Nishasta. One involves soaking the pre-washed wheat grains for one or two days, followed by grinding and straining or decanting the resultant mixture to remove the bran, and finally drying. The other method follows a traditional procedure that involves soaking of pre-washed wheat grains in water for few weeks in a closed container to allow natural fermentation to take place. The water is changed every week to allow fresh microbial growth. This is followed by crushing of the grains by hand and filtering the residue through a muslin cloth and drying [13]. The latter method effectively allows the fermentation to takes place and was followed in the current study for the preparation of the samples. The current study aimed to characterize the microbial composition of Nishasta and to explore its effect in retinoic acid-induced osteoporosis in female Wistar rats.

Fructans constitute the predominant component of the whole wheat grains in the form of inulin and other oligofructose derivatives. They represent the non-digestible polymers of fructose with a mean degree of polymerization of 25. This non-digestible portion of wheat forms the fermentable dietary fiber (prebiotics). Microbial fermentation of wheat results in the formation of partially hydrolyzed products such as short-chain fructo-oligosaccharides. Their effects on calcium absorption and bone mineral density are well established [22,23]. The wheat bran also contains phytic acid and its derivatives that have been shown to decrease the bioavailability of calcium, thereby acting as anti-nutrients. Fermentation serves the additional purpose of decreasing the content of phytic acid derivatives from the fermentation milieu.

The bacterial isolates from the fermented wheat were examined based on culture characteristics and macroscopic analysis. The results of microbial characterization revealed the presence of ten clones of *Lactobacillus plantarum* in the fermented preparation with *L. plantarum* NF3 as the predominant strain. *L. plantarum* is a facultative lactic acid bacterium that is mainly found in the fermentation of plant-derived raw materials. The average microbial count was $2.4 \times 10^3$ CFU/g of sample. The study established that Nishasta contains a bolus of useful microbes and thereby can act as a probiotic.

A systemic bone disorder that results in diminished bone mass, increased risk of fractures, and altered micro-structure of bones is called osteoporosis. Several plant constituents have been reported to be beneficial in osteoporosis in experimental animals. These include soy isoflavones [24,25], quercetin [26], naringin [27], and diosgenin [28]. Fermented *Moringa oleifera* has been reported to be beneficial against bone loss [29] and hepatic adiposity and glucose intolerance [30]. This study aimed to evaluate the effect of feeding fermented wheat to Wistar rats with retinoic acid-induced osteoporosis. Retinoic acid administration is known to damage the function of ovaries and reduce the estrogen level, thereby resulting in osteoporosis. This in turn activates osteoclasts and increases bone absorption [31].

Bone loss occurs when the extent of bone resorption is greater than bone formation. Several bone biochemical markers such as ALP and TRACP allow a dynamic evaluation of the bone-curative effect of the treatment regimens. ALP and TRACP are the prime biomarkers predicting bone differentiation and bone formation. These markers are upregulated under osteoporotic conditions [32]. The serum calcium and phosphorus levels showed marked increases in the fermented wheat-treated groups compared with the model. Levels of serum ALP and TRACP decreased significantly in fermented wheat-treated groups compared to the model. The restorative effect of wheat treatment was comparable to sodium alendronate. Fermented wheat successfully truncated the increased levels of serum TRACP and ALP, which indicated that it inhibited bone resorption and improved bone differentiation in retinoic acid-induced osteoporosis in Wistar rats. The excretion of calcium, phosphorus, and HOP in urine increased significantly after retinoic acid was administered for 14 days compared with the healthy control. The excretion of calcium, phosphorus, and HOP in urine decreased significantly in fermented wheat-treated animals compared with the model.

From the results, it is quite clear that the bone weight index in the model was significantly lower than that in the healthy control ($p < 0.05$). Upon treatment with the high dose of fermented wheat, the bone weight index improved significantly compared with the model ($p < 0.05$). The index was $0.38 \pm 0.01$ with a high dose compared to $0.29 \pm 0.02$ in the model. The femoral head diameter and length of the femur increased after treatment with fermented wheat. The effect of the high dose of fermented wheat was particularly comparable to sodium alendronate. Retinoic acid significantly decreased the ash content of the femur bone when compared with that of healthy rats ($p < 0.05$). The bone ash content in fermented wheat-treated animals showed a marked improvement ($34.85 \pm 0.21$ g/100 g for animals administered a high dose of fermented wheat compared to $27.43 \pm 0.15$ g/100 g for the model). A dose-dependent effect of fermented wheat in improving the bone parameters was generally observed. Upon administration of retinoic acid, bone calcium and phosphorus levels decreased significantly. Treatment with fermented wheat markedly improved the bone calcium and phosphorus contents compared with the model. Fermented wheat did not exhibit a dose-dependent effect in increasing the bone calcium and phosphorus contents. Treatment with sodium alendronate was able to restore their contents significantly ($p < 0.001$).

Calcium and phosphorus represent the main mineral components of bone tissues. These are also essential in maintaining bone density. Retinoic acid-induced osteoporotic animals exhibited reduced levels of serum calcium and phosphorus. An increase in osteoclasts in osteoporosis promotes bone resorption and enhances bone mineral dissolution. Subsequently, calcium and phosphorus levels decrease in bone and increase in urine. Increased urinary calcium excretion suggested increased bone resorption. Administration of fermented wheat reduced the levels of urinary calcium and phosphorus, indicating a resumption of mineral homoeostasis. Salts of calcium are incorporated in collagen fibrils of bones [33]. Breakdown of collagen fibrils occurs during bone loss, resulting in the appearance of HOP in urine. An increased content of HOP indicates more breakdown of collagen and it is considered as an index of bone absorption [34,35]. It was found that the fermented wheat substantially decreased the excretion of HOP and thus prevented bone deterioration. Decreased levels of phosphorus in urine may be due to its enhanced transport across the border membrane. Treatment with fermented wheat also produced a substantial increase in the phosphorus content of bones.

Gross changes in the micro-structure and BMD of the femur of rats after retinoic acid administration were also observed. Treatment with fermented wheat (200 mg/kg b.w., HD-FW) alleviated the effects of retinoic acid and helped in maintaining and rebuilding the bone microstructure. It also showed enhancement in the trabecular volume percentage, thickness, and number, as well as a decrease in trabecular separation compared with the model. Administration of retinoic acid to rats produced significant decreases in femoral BMD that were significantly reversed after treatment with fermented wheat in a dose-dependent

manner compared to the model. This also increased bone mass and mineral density. The results showed that fermented wheat had a restorative effect against osteoporosis. The effect was similar to that of sodium alendronate. Treatment with fermented wheat prevented the loss of bone calcium and phosphorus and improved BMD in osteoporotic rats. Improvement in bone health parameters with fermented wheat was in agreement with the decrease in urine calcium and phosphorus contents in treated groups. Moreover, treatment with fermented wheat increased the length of the femur and diameter of its head. The bone weight index also increased significantly. Similarly, the dimensions of the tibia head and length were normalized by fermented wheat treatment.

The results also demonstrated that the fermented wheat-treated rats had better bone calcium, phosphorus, and ash contents compared with the model. The bone mass restoring effect of fermented wheat in osteoporotic rats was evident. It is well known that there is a decrease in bone weight during osteoporosis, and the same was seen in rats administered retinoic acid. The bone weight of the retinoic acid-treated rats was lower than that of the healthy control. Upon treatment with fermented wheat, the bone weight index of the rats returned to normal. These results indicated that fermented wheat played an active role in enhancing the overall bone quality. Our results are in agreement with the reports advocating the use of dietary components to improve bone quality [36,37]. It is noteworthy that the treatment with fermented wheat significantly ameliorated the changes induced by retinoic acid in vital organs (See Supplementary Material, Figure S1). The body weight of the rats also decreased during induction of osteoporosis in all groups except the healthy control. The trend was reversed after treatment with fermented wheat. The restorative effect was more pronounced with the high dose of fermented wheat (200 mg/kg) than the medium and low doses.

One of the possible explanations for the beneficial effects of Nishasta in osteoporosis is the enhanced production of short-chain fatty acids (SCFAs) by providing suitable substrates in the gut (prebiotic). It also adds to the gut microbiotic composition by providing homologs of *Lactobacillus plantarum* (probiotic). Microbial fermentation and hydrolysis reactions transform prebiotics to SCFAs, thereby lowering the pH of the intestinal luminal contents [38]. Greater acidity in the colon is known to release calcium from negatively charged metabolites such as phytates, which in turn increases its availability for absorption and subsequent bone mineralization. Prebiotic fibers have been shown to increase the intestinal content of SCFAs in animal models [39–41]. Moreover, *L. plantarum* has been reported to prevent retinoic acid-induced osteoporosis in rats [42]. On the basis of the foregoing account, Nishasta is postulated to prevent the bone destruction in retinoic acid-induced osteoporosis through prebiotic- and probiotic-assisted bone mineralization.

## 5. Conclusions

The fermentation improved the prebiotic potential of wheat. *Lactobacillus plantarum* was the predominant microbe in the fermented samples and provided the probiotic bolus. The current study clearly demonstrated the bone quality improving effect of Nishasta in retinoic acid-induced osteoporosis in rats. Its beneficial effect was established on the basis of biochemical, physicochemical, and histological evidence. In conclusion, the study established the anti-osteoporosis activity of Nishasta in Wistar rats that can be partly attributed to the improved gut calcium absorption and microbiota composition due to the fermented preparation. Identification and evaluation of metabolites synthesized during the fermentation of wheat can be the prospects for future research.

**Supplementary Materials:** The following supporting information can be downloaded at: https://www.mdpi.com/article/10.3390/fermentation8040182/s1. Figure S1: Effect of fermented wheat on histology of vital organs. All the slides were photographed at 10× and 40× magnifications.

**Author Contributions:** Investigation, A.A.; carried out formal analysis, N.H.A. and Z.U.; designed and supervised the work, B.P.P.; curated the data, S.A.; conceptualized and designed the work, S.R.M. All authors have read and agreed to the published version of the manuscript.

**Funding:** The research work was funded by the Indian Council of Medical Research (ICMR), New Delhi Grant No. 3/1/2/153/2019-(Nut).

**Institutional Review Board Statement:** Not applicable.

**Informed Consent Statement:** Not applicable.

**Data Availability Statement:** Not applicable.

**Acknowledgments:** The authors would like to express their sincere thanks to Divya Singh, Department of Endocrinology, Central Drug Research Institute (CDRI), Lucknow, India for carrying out BMD assay. We are also thankful to Divya Vohora, Department of Pharmacology, School of Pharmaceutical Education & Research, Jamia Hamdard, New Delhi, for her support. The authors sincerely acknowledge the Department of Pharmacognosy & Phytochemistry, School of Pharmaceutical Education & Research, Jamia Hamdard, New Delhi, for the infrastructural support developed under UGC-SAP DRS-II (No.F.3-19/2015/DRS-II SAP-II).

**Conflicts of Interest:** The authors declare that they have no known competing financial interests or personal relationships that could have appeared to influence the work reported in this paper.

## Abbreviations

| | |
|---|---|
| ALP | alkaline phosphatase |
| BMD | bone mineral density |
| CMC | Carboxymethyl cellulose |
| HC | Healthy control |
| HD-FW | High-dose fermented wheat |
| HOP | hydroxyproline |
| LD-FW | Low-dose fermented wheat |
| Micro-CT | microtomography |
| MD-FW | medium dose fermented wheat |
| MRS | DeMan–Rogosa–Sharpe |
| PCR | polymerase chain reaction |
| SCFAs | Short-Chain Fatty Acids |
| TRACP | tartrate resistant alkaline phosphatase. |

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
