# Peer review of "Microbial Composition of a Traditional Fermented Wheat Preparation—Nishasta and Its Role in the Amelioration of Retinoic Acid-Induced Osteoporosis in Rats"

_fermentation, doi:10.3390/fermentation8040182_

Round 1

Reviewer 1 Report

The manuscript fermentation-1635600 analyses the role of fermented wheat (Nishasta, Indian traditional product) in contrasting retinoic-acid induced osteoporosis (OP) in mouse. Due to the growing interest on the potential effects of fermented products for human health health, this work might be of interest for the journal’s readers. However, I believe that the results need a clearer presentation, and the following issues must be addressed to bring the paper to a publishable standard, as follows.

Major comments

2.2 Method of fermentation. It seems that a single fermentation batch was obtained and analysed here. How can the authors assess the reproducibility of their results? Especially considering the fact that the fermentation proceeds spontaneously with no microbial inoculation

2.3 Isolation and molecular characterization of microbes. How many colonies were obtained and how many analysed? were they representative of the microbial population? Also, there is no mention of the primers used. More importantly, the results of this analysis are not presented (e.g., agarose gel and identification/similarity search outputs). This is a serious flaw considering that the authors state “The main objective of the study was to characterize the microbial composition of Nishasta” (lines 74-75)

2.4 Animals and experimental design. The use of the label “negative control” to identify mice treated with retinoic acid to induce OP can be misleading. Please indicate this group as “model” or “OP model” for the sake of clarity. The same applies to the following sections, graphs, and tables as appropriate. Also, the authors should justify the dose of sodium alendronate (40 mg/kg) used here given the fact that other papers report far lower dosages for anti-osteoporotic activity (2.5 – 10 mg/kg). Similarly, authors should provide an explanation for the dosages (high, medium and low) of fermented wheat they tested on OP mice.

Results.

One of the major drawbacks of this manuscript is the lack of clarity in how results are presented.

It seems that Table 1 reports the parameters already presented in the manuscript Figures: body weight (Fig. 1); blood calcium, phosphate, ALP and TRACP (Fig. 2); urine calcium, phosphate, and HOP (Fig. 3); bone parameters (Fig. 4); bone minerals (Fig. 5). What is the point of replicating such information?

Furthermore, the authors should consider splitting their data according to the main goal, i.e., validation of the OP model (14 days) and effects of treatments (indicated as 56 days in Figures, 42 days in the text – this requires uniformity too). This would increase the readability of their results.

Figure legends must be implemented to be self-explanatory.

RANKL and OPG should be investigated too.

Discussion

For the most part, the Discussion is a repetition of what already presented in the Introduction or in the Results section. The only valuable part is the last paragraph. Authors are strongly encouraged to address this issue and to mitigate their comments since they are not able to provide any results corroborating the postulated mechanisms of action of Nishasta as an anti-osteoporotic agent

Minor comments

Lines 59-60 need reference(s)

General comments

There are mistakes and typos throughout the manuscript

Author Response

Please find attached the compliance to reviewers comments/suggestions.

Reviewer 2 Report

Overall, this is an interesting study and a well prepared manuscript. Some comments and suggestions are provided.

  1. How to scale up the proposed approach in this study?
  2. How to avoid the contamination of miscellaneous bacteria?
  3. How about the economical feasibility of the proposed technology?
  4. A schematic diagram of the entire fermentation setup is suggested to be supplemented in the main body of the manuscript. That would be helpful to the readers to understand the process. 

Author Response

(The authors gave the same response as above.)

Reviewer 3 Report

Dear authors, I have read with interest the manuscript entitled "Microbial Composition of a Traditional Fermented Wheat Preparation - Nishasta and its Role in Amelioration of Retinoic Acid Induced Osteoporosis in Rats". The research conducted by the authors is interesting and brings new information regarding the microbial composition and importance in food preparation.

There are some suggestions and comments that can improve the work of the authors.

Abstract - make a condensed form from the sentences that belong to Material and Methods (lines 16-23). In this way your abstract will be more focused on results

-Keywords - I suggest you to remove the words that are present in title too. Change them with similar ones. 

Introduction - this section must be improved. You have only 7 references. You need to expand the entire text of this section and connect it to international literature, and with similar works.

The last paragraph of the introduction should be separated from the text and to provide a clear aim of your work and objectives/hypotheses, clear separated one from others (e.g. i), ii) or a), b) or 1), 2) ...).

Pay attention (in the entire text) to the formating (line 70-74).

Material and Methods - describe in a condensed form the preparation of Nishasta. You have a description in the introduction, but it should be moved in this section.

Results section - This section is very important for your article. It should be organized to present at the maximum your results and observations recorded during the experiment. Please expand the interpretation of your results in sub-section 3.1. Proximate analysis. 3.2. - This subsection present methodological aspects rather than results. In this form is too short for a Results sub-section. 3.3. expand it. 3.4. - must expand it, you have multiple figures that can be analyzed. 3.5. expand it. 3.6, 3.7 - expand the interpretation.

Discussion section - do not repeat the same text as in the introduction - lines 342-346. Table 1 should be moved with its text to Results section. Also, any text that present results, not discussion of the results, should be moved in the Results section. Do not make connections with previous presented figures/tables or insert tables/figures in this section. Here you need to compare your results (as a discussion) with international references in the field.

 Conclusion section - add values of your main results.

the overall article is interesting and deserve improvements in order to better present your results.

Author Response

(The authors gave the same response as above.)

Round 2

Reviewer 1 Report

The authors have addressed adequately my previous comments except the following:

2.4) it seems that they did not include the citation supporting the tested dosages of fermented foods (available in the rebuttal letter)

results) they did not provide an explanation about why they chose not to investigate proteins involved in bone regulation (e.g., RANKL and OPG)

Author Response

Compliance to Reviewers’ Comments/Suggestions (R2)

Reviewer 1:

Comments and Suggestions for Authors

The authors have addressed adequately my previous comments except the following:

Section 2.4) it seems that they did not include the citation supporting the tested dosages of fermented foods (available in the rebuttal letter)

Results) they did not provide an explanation about why they chose not to investigate proteins involved in bone regulation (e.g., RANKL and OPG)

Response/Compliance:

The manuscript has been revised as per the suggestions of the review and the said reference has been included in the Section 2.4 at #[16].

The authors are unable to carry out RANKL and OPG analyses because the experimental procedure is already completed. However, the authors have noted the suggestion for future.

Reviewer 2 Report

The authors have addressed all the comments as per my review comments.

Author Response

The manuscript was revised as per the suggestions.

Reviewer 3 Report

Dear authors, the article looks better in this form. There is still place for expanding the results section if you consider.

Author Response

Compliance to Reviewers’ Comments/Suggestions (R2)

Reviewer 3:

Comments and Suggestions for Authors

Dear authors, the article looks better in this form. There is still place for expanding the results section if you consider.

Response/Compliance:

The manuscript has been revised as per the suggestions of the review.

Relevant details have been elaborated wherever necessary. Repeating the explanations for the data presented in the form of figures has been intentionally avoided.